# Impact of Workplace Bullying on Quiet Quitting in Nurses: The Mediating Effect of Coping Strategies

**DOI:** 10.3390/healthcare12070797

**Published:** 2024-04-06

**Authors:** Petros Galanis, Ioannis Moisoglou, Aglaia Katsiroumpa, Maria Malliarou, Irene Vraka, Parisis Gallos, Maria Kalogeropoulou, Ioanna V. Papathanasiou

**Affiliations:** 1Clinical Epidemiology Laboratory, Faculty of Nursing, National and Kapodistrian University of Athens, 11527 Athens, Greece; aglaiakat@nurs.uoa.gr (A.K.); parisgallos@nurs.uoa.gr (P.G.); mariakalo@nurs.uoa.gr (M.K.); 2Faculty of Nursing, University of Thessaly, 41500 Larissa, Greece; iomoysoglou@uth.gr (I.M.); malliarou@uth.gr (M.M.); iopapathanasiou@uth.gr (I.V.P.); 3Department of Radiology, P. & A. Kyriakou Children’s Hospital, 11527 Athens, Greece; irenevraka@yahoo.gr

**Keywords:** bullying, quiet quitting, nurses, workplace, coping strategies, mediation analysis

## Abstract

Workplace bullying is common among nurses and negatively affects several work-related variables, such as job burnout and job satisfaction. However, no study until now has examined the impact of workplace bullying on quiet quitting among nurses. Thus, our aim was to examine the direct effect of workplace bullying on quiet quitting and to investigate the mediating effect of coping strategies on the relationship between workplace bullying and quiet quitting in nurses. We conducted a cross-sectional study with a convenience sample of 650 nurses in Greece. We collected our data in February 2024. We used the Negative Acts Questionnaire—Revised, the Quiet Quitting Scale, and the Brief COPE to measure workplace bullying, quiet quitting, and coping strategies, respectively. We found that workplace bullying and negative coping strategies were positive predictors of quiet quitting, while positive coping strategies were negative predictors of quiet quitting. Our mediation analysis showed that positive and negative coping strategies partially mediated the relationship between workplace bullying and quiet quitting. In particular, positive coping strategies caused competitive mediation, while negative coping strategies caused complimentary mediation. Nurses’ managers and policy makers should improve working conditions by reducing workplace bullying and strengthening positive coping strategies among nurses.

## 1. Introduction

Incidents of violence are a frequently occurring phenomenon in the working life of nurses. Workplace bullying is a type of violence, which can take the form of persistent negative mistreatment, consisting of frequent and constant criticism and person-related physical, verbal, or psychological violence [1,2]. Any conflict or confrontation in the workplace does not constitute bullying, which has certain defining characteristics. These include the fact that the employee is systematically targeted (by peers, superiors, or even subordinates) by becoming the subject of negative and unwanted social behavior in the workplace; this targeting lasts for a long period of time and the victim of this behavior can neither easily escape the situation nor stop the unwanted treatment [3]. It is estimated that at least one in ten healthcare professionals is a victim of workplace bullying [4,5], while the prevalence among nurses varies between 2.4% and 94% [6,7]. A systematic review of qualitative studies highlighted the “faces” of nurses’ bullying, where they indicated that they experienced being excluded and isolated, facing verbal abuse and hostility, and being excessively scrutinized, silenced, oppressed, and threatened by those in power. They also reported being trapped in a system of intimidation, being provided with limited career opportunities, and having their reputation damaged [8].

The factors that trigger the occurrence of bullying in the working environment of nurses are mostly related to organizational issues. In particular, the work characteristics of the nursing profession, such as work overload, shift work, job demands, severe staff shortages, and stress, were found to be associated with the development of bullying behavior [9,10]. Also, supervisors lack adequate skills to manage bullying incidents and support victims; in addition, lack of organizational support and support from colleagues, disruptive working relationships, tolerance of bullying incidents, lack of policies on bullying, and lack of prevention measures are some of the most important antecedents for the occurrence of bullying incidents among nursing staff [9,10]. The effects of bullying are multidimensional and affect nurses, patients, and the functioning of the organization. Nurses who are bullied are more likely to experience stress, burnout, job dissatisfaction, depression, anxiety, post-traumatic stress disorder, low self-esteem, physical health symptoms, and deterioration in the quality of their work life [9,11,12]. Patients who are hospitalized in departments where nursing staff experience bullying may experience errors and adverse events and may not receive comprehensive nursing care [13,14,15]. At the organizational level, nurses’ relationships are disrupted as barriers to teamwork and communication develop due to bullying incidents [13]. Also, high rates of absenteeism of nurses from work are recorded; they have a reduced commitment to the organization and report their turnover intention [6,16,17].

In the case of bullying incidents, the reaction of the victim is crucial, especially the reaction of their management. Studies show that nurses use both positive and negative management strategies [18,19,20]. The positive ones include being problem-focused, seeking social support, and having crucial conversations. The negative ones, which are often the most common reactions, include wishful thinking, detachment, evasion, substance use, and leaving the organization. By choosing avoidance and wishful thinking, no reduction in bullying incidents is achieved; the problem remains and will continue to manifest. The other negative strategies actually harm the nurses (substance use) and have negative consequences for the functioning of the healthcare organization (leaving the organization). The above makes it imperative to train nursing staff in the optimal management of bullying incidents, as well as to establish organizational policies of zero tolerance to violence regardless of where it comes from.

During the COVID-19 pandemic, a new phenomenon emerged among employees, that of quiet quitting, which was presented on the video creation and sharing platform, TikTok [21]. Employees who opt for quiet quitting do not resign from their job or their professions but reduce their performance. Specifically, they perform the minimum requirements of their job, barely enough to avoid being fired, do not express new ideas, do not stay overtime, and do not arrive at work earlier than the designated arrival time [21,22]. Although some argue that this is an old phenomenon [23], until recently there was no literature on the extent of the issue, the factors that cause it, and its impact. A large study in the business sector in the US during the COVID-19 pandemic by Gallup showed that half of the employees are quiet quitters [24]. In the same period, a study of the phenomenon was also initiated in the healthcare sector, with the development of a reliable and valid tool for measuring quiet quitting [25], which revealed that in this sector too, more than 50% of employees opt for quiet quitting, highlighting the urgency of the issue [26]. In particular, nurses had the highest rates of quiet quitting (67.4%) compared to other healthcare professionals [26]. Studies have shown that burnout is a predictor of quiet quitting [27], which in turn increases the likelihood of turnover intention among nurses [28], while moral resilience negatively influences the occurrence of quiet quitting [29]. To the best of our knowledge, no study until now has examined the impact of workplace bullying on quiet quitting in nurses. Thus, our first hypothesis was the following:

**H1.** 
*Workplace bullying would have a direct effect on quiet quitting in nurses. In other words, we hypothesized that the higher the levels of workplace bullying, the higher the quiet quitting in nurses.*


The highly demanding nursing profession often confronts nurses with stressful situations that affect their physical and mental health. Faced with these situations, regardless of the degree of organizational support, nurses are required to cope with them in order to reduce their negative impact. Coping with a stressful situation is the third part of a procedure, where the primary appraisal is the process of perceiving a threat to oneself, the secondary appraisal is the process of bringing to mind a potential response to the threat, and coping is the process of executing that response [30]. Studies have shown the beneficial effect of coping strategies on nurses in reducing their levels of stress, burnout, and compassion fatigue and enhancing their psychological well-being [31,32,33,34]. Considering the essential role of coping strategies as mediators in studies including nurses, we examined the following second hypothesis:

**H2.** 
*Coping strategies would be a mediator in the relationship between workplace bullying and quiet quitting in nurses. In other words, we hypothesized that nurses who received more workplace bullying may employ more maladaptive (or negative) coping strategies (e.g., self-blame) and less adaptive (or positive) coping strategies (e.g., active coping), and therefore experience higher levels of quiet quitting.*


In short, our aim was to examine the direct effect of workplace bullying on quiet quitting and to investigate the mediating effect of coping strategies on the relationship between workplace bullying and quiet quitting in nurses (Figure 1).

## 2. Materials and Methods

### 2.1. Study Design

We conducted a web-based cross-sectional study with nurses in Greece. We used Google forms to create an online version of the study questionnaire. Then, we distributed the questionnaire through nursing groups on Facebook, LinkedIn, Viber, and WhatsApp. Thus, a convenience sample was obtained. Our inclusion criteria were the following: (a) nurses who have been working in clinical settings, such as hospitals and healthcare centers, (b) nurses who have been working for at least two years in order to experience workplace bullying, and (c) nurses who understand the Greek language. We collected our data in February 2024.

### 2.2. Measures

We used the Negative Acts Questionnaire—Revised (NAQ-R) to measure workplace bullying among nurses [35]. The NAQ-R consists of 22 items measuring work-related bullying, person-related bullying, and physically intimidating bullying during the last six months. Answers are on a 5-point Likert scale: never (1), now and then (2), monthly (3), weekly (4), and daily (5). A total score from 22 to 110 is obtained by summing up all answers. Higher scores on NAQ-R indicate higher levels of workplace bullying. We used the valid Greek version of the NAQ-R [36]. In particular, Kakoulakis et al. [36] validated the Greek version of the NAQ-R in a sample of teachers by investigating the reliability and validity of the tool. In that study, Cronbach’s alpha for the NAQ-R was 0.83, while the concurrent validity of the tool was high since scholars found statistically significant correlations between NAQ-R and self-esteem (r = −0.364), stress (r = 0.406), and depression (r = 0.389). In our study, Cronbach’s alpha for the NAQ-R was 0.963.

We used the Quiet Quitting Scale (QQS) to measure the levels of quiet quitting among our nurses [25]. The Greek version of the QQS has been validated in a sample of nurses in Greece [26]. The QQS consists of nine items measuring detachment, lack of initiative, and lack of motivation. Answers are on a 5-point Likert scale: strongly disagree/never (1), disagree/rarely (2), neither disagree or agree/sometimes (3), agree/often (4), and strongly agree/always (5). Answers to nine items are averaged to compose an overall score on QQS. Overall QQS score takes values from 1 (low levels of quiet quitting) to 5 (high levels of quiet quitting). Developers of the QQS suggest a cut-off point of 2.06 to discriminate quiet quitters from non-quiet quitters [37]. In that study, researchers used a sample of workers from every job sector to identify a cut-off point for the QQS. In particular, they used several external criteria (i.e., the “Job Satisfaction Survey”, the “Copenhagen Burnout Inventory”, and the “Single Item Burnout Measure”) and they performed the Receiver Operating Characteristic analysis to estimate the best cut-off point for the scale. In our study, Cronbach’s alpha for the QQS was 0.816.

We used the Brief COPE to measure coping strategies in our sample [38]. The Brief COPE consists of 28 items measuring the following 14 dimensions: self-distraction, active coping, denial, substance use, use of emotional support, use of instrumental support, behavioral disengagement, venting, positive reframing, planning, humor, acceptance, religion, and self-blame. Answers are on a 4-point Likert scale: I have not been doing this at all (1), I have been doing this a little bit (2), I have been doing this a medium amount (3), and I have been doing this a lot (4). The score ranges from 1 to 4. Higher values indicate a higher adaptation of coping strategy. We used the valid Greek version of the Brief COPE [39]. In this study, scholars used a sample of Greek-speaking adults and found adequate psychometric properties for the Greek version of the scale. In particular, scholars performed exploratory and confirmatory factor analysis, and they calculated Cronbach’s alpha for the factors. In our study, all Cronbach’s alphas for the 14 dimensions were above 0.60. A recent systematic review revealed that most studies used the Brief COPE have identified a two-factor structure: approach or positive coping strategies (active coping, use of emotional support, use of instrumental support, positive reframing, planning, acceptance, religion, venting, and humor) and avoidance or negative coping strategies (self-distraction, denial, self-blame, behavioral disengagement, and substance use) [40]. Thus, we followed this two-factor structure in our study. In other words, approach coping is considered a positive/adaptive/engaged/active/direct strategy, while avoidance coping is considered a negative/maladaptive/disengaged/indirect strategy. Scores on positive and negative coping strategies range from 1 (low adaptation of strategy) to 4 (high adaptation of strategy).

We considered gender (females/males), age (continuous variable), understaffed department (no/yes), clinical experience (continuous variable), and shift work (no/yes) as covariates in the mediation models.

### 2.3. Ethical Issues

We informed nurses about the aim and the design of the study and obtained their informed consent to participate. We did not collect personal data. We followed the guidelines of the Declaration of Helsinki in our study. The study protocol was approved by the Ethics Committee of the Faculty of Nursing, National and Kapodistrian University of Athens (approval number; 479, 10 January 2024).

### 2.4. Statistical Analysis

As we explained above, we considered positive (approach) coping and negative (avoidance) coping as potential mediators in the relationship between workplace bullying (independent variable) and quiet quitting (dependent variable). Hair et al. suggest that the number of participants should be at least 10 times that of the study variables in the mediation analysis [41]. Since the NAQ-R (predictor variable) consists of 22 items, the Brief COPE (mediator variable) consists of 28 items, and the number of covariates was five; the required sample size was 550 nurses (=55 variables × 10 = 550).

We present categorical variables with numbers and percentages. Moreover, we present continuous variables with mean, standard deviation, median, minimum value, maximum value, and range. We calculated Pearson’s correlation coefficient to determine the correlation between workplace bullying, quiet quitting, positive coping strategies, and negative coping strategies. We constructed a multivariable linear regression model to determine the independent effect of workplace bullying on quiet quitting. In that case, we eliminated the confounding caused by demographic and job characteristics of nurses. The correlation between age and clinical experience was very high (r = 0.912, *p* < 0.001). Thus, to avoid collinearity in the multivariable linear regression model, we decided to include one variable (age) in our model. We present adjusted coefficients beta, 95% confidence intervals (CI), and *p*-values for variables in the multivariable linear regression model.

We used the PROCESS macro (Model 4) [42] to test the mediating effect of positive and negative coping strategies in the relationship between workplace bullying and quiet quitting. We based our mediation analysis on 5000 bootstrap samples [43]. We calculated the 95% Cis, regression coefficients (b), and standard errors. The *p*-values less than 0.05 were considered statistically significant. We used IBM SPSS 21.0 (IBM Corp. Released 2012; IBM SPSS Statistics for Windows, Version 21.0.: IBM Corp., Armonk, NY, USA) for statistical analysis.

## 3. Results

### 3.1. Demographic and Job Characteristics

Our sample included 665 nurses. The mean age of nurses was 38.9 years (SD = 10.1), while the median and range were 39 and 42 years, respectively. The majority of nurses were females (87.7%). Among our nurses, 80.2% stated that they have been working in understaffed departments and 74.4% were shift workers. The mean years of clinical experience were 14.1 (SD = 10.2), the median value was 14 years, and the range was 39 years. Table 1 shows the detailed demographic and job characteristics of our nurses.

### 3.2. Study Scales

Descriptive statistics for the study scales are shown in Table 2. The mean value of the NAQ-R was 51.3 (SD = 20.6), while the mean value of the QQS was 2.5 (SD = 0.6). Applying the cut-off point (2.06) for the QQS, we found that 77.3% (*n* = 514) of our nurses were quiet quitters, while 22.7% (*n* = 151) were non-quiet quitters. Nurses employed positive coping strategies more often than negative coping strategies, since the mean scores were 2.6 (SD = 0.5) and 2.0 (SD = 0.5), respectively.

### 3.3. Correlation Analysis

Table 3 shows Pearson’s correlation analysis of workplace bullying, quiet quitting, positive coping strategies, and negative coping strategies. We found a positive correlation between workplace bullying and quiet quitting (r = 0.453, *p* < 0.001), positive coping strategies (r = 0.244, *p* < 0.001), and negative coping strategies (r = 0.423, *p* < 0.001). Moreover, we found a positive correlation between negative coping strategies and quiet quitting (r = 0.395, *p* < 0.001).

### 3.4. Regression Analysis

We conducted a multivariable linear regression analysis to examine Hypothesis 1. We found that workplace bullying had an independent positive effect on quiet quitting (adjusted beta = 0.010, 95% CI = 0.008 to 0.012, *p* < 0.001). Workplace bullying explained 20.4% of the variance of quiet quitting. Therefore, Hypothesis 1 was proved. Gender, positive coping strategies, and negative coping strategies were also associated with quiet quitting. These three independent variables explained 7.7% of the variance of quiet quitting. Analysis of variance for the multivariable model was statistically significant (*p* < 0.001). Table 4 shows the detailed results from the multivariable linear regression analysis.

### 3.5. Mediation Analysis

Table 5 shows the indirect impact of workplace bullying on quiet quitting through positive and negative coping strategies. Our mediation analysis showed that the indirect mediated effect of workplace bullying on quiet quitting through positive coping strategies was significant (b = −0.0011, 95% CI = −0.0017 to −0.0005, *p* < 0.0001). Workplace bullying was a significantly positive predictor of quiet quitting (b = 0.0128, 95% CI = 0.0109 to 0.0147, *p* < 0.0001). Workplace bullying was a significantly positive predictor of positive coping strategies (b = 0.0060, 95% CI = 0.0042 to 0.0078, *p* < 0.0001), while positive coping strategies were a significantly negative predictor of quiet quitting (b = −0.1754, 95% CI = −0.2574 to −0.0933, *p* < 0.0001). Additionally, workplace bullying was a significantly positive predictor of negative coping strategies (b = 0.0094, 95% CI = 0.0079 to 0.0110, *p* < 0.0001), while negative coping strategies were a significantly positive predictor of quiet quitting (b = 0.3864, 95% CI = 0.2891 to 0.4837, *p* < 0.0001). Moreover, the direct effect of workplace bullying on quiet quitting was still significant (b = 0.0103, 95% CI = 0.0082 to 0.0122, *p* < 0.0001) even after the mediating effect of positive and negative coping strategies. Positive and negative coping strategies partially mediated the relationship between workplace bullying and quiet quitting since the direct and indirect effects of workplace bullying were significant. Positive coping strategies caused partial competitive mediation, while negative coping strategies caused partial complementary mediation. In conclusion, our mediation analysis supported Hypothesis 2. Figure 2 presents the final mediation model.

## 4. Discussion

The present study revealed that workplace bullying and negative coping strategies were positive predictors of quiet quitting, while positive coping strategies were negative predictors of quiet quitting. The mediation analysis showed that positive and negative coping strategies partially mediated the relationship between workplace bullying and quiet quitting.

The present study is the first to highlight the association between bullying and quiet quitting among nurses. We found that 77.3% of our participants can be considered quiet quitters. The high incidence of bullying in healthcare organizations may exacerbate the phenomenon of quiet quitting, which in any case seems to be widespread. Nurses who opt for quiet quitting are actually giving an image of adequate staffing, where underneath the numerical staffing lies reduced performance, lack of creativity, and innovative behavior. Nurses are caught in the middle of a constant tug-of-war, where on one side are the high job demands and requirements to improve the outcomes of healthcare organizations, and on the other side is the long-standing inability of healthcare organizations to ensure adequate resources and organizational support needed in the challenging task of providing healthcare by nurses. As a result of the above, nurses experience high rates of burnout, dissatisfaction, stress, and depression. Bullying is another burdening factor in the work environment of nurses, as it becomes a source of burnout, depression, psycho-physical consequences, turnover, and leaving the nursing profession [44,45,46]. By opting for quiet quitting, employees are essentially trying to balance their personal and work lives [21]. A study by the Harvard Business Review in the field of business showed that as a manager’s ability to balance between achieving results and caring about others increases, quiet quitting decreases and employees’ willingness to put in more effort increases [47]. Similar findings were observed in the health sector regarding bullying, where nurse managers with a low relationship-oriented leadership style are associated with the occurrence of bullying and turnover intention [48]. When the nurse manager chooses caring leadership, it reduces the likelihood of their staff being exposed to bullying behavior [49]. Also, when the victims of bullying receive organizational support, their satisfaction increases and absenteeism from work decreases [50].

Regardless of the manager’s response to the bullying and the degree of organizational support, the nurse is also required to cope with the stressful situation of the bullying episode in which they are involved. The results of the present study showed that nurses choose more positive bullying management strategies compared to negative ones. These findings are consistent with those of other studies, where positive strategies also scored higher [19,20]. However, studies show that there is great diversity in the bullying coping strategies chosen by nurses. A cross-cultural scoping review showed that nurses used emotion-focused coping strategies more frequently almost in all clusters [9]. In a study involving nurses from Portugal, negative bullying coping strategies including substance use and resorting to evasion predominated, and in their majority, nurses did not receive training on bullying management [20]. In contrast, nurses in Australia seem to choose positive coping strategies such as being problem-focused and seeking social support [51]. In another study, the nurses were more likely to report distraction, substance use, emotional support, disengagement, venting, positive reframing, humor, and religion [52]. Therefore, an effective attempt to address the problem is to provide nurses with the skills and guidance needed to deal with bullying, using positive coping strategies such as conflict management and assertiveness [53].

As bullying is a negative and stressful incident in the workplace, the choice of positive coping strategies by the victim is an effective factor in mitigating the negative effects of the incident and contributing to the well-being of the victim [54]. When nurses apply positive bullying coping strategies, such as more approach-oriented strategies and fewer avoidance-oriented strategies, these strategies are found to be associated with greater psychological well-being and fewer mood disturbances [33,55]. Psychological well-being in turn was directly related to the quality of nurses’ practice environment and safety attitudes [33]. As bullying deteriorates the quality of nurses’ working lives, the adoption of positive bullying coping strategies moderates this negative effect [56]. Another effective bullying coping strategy that can be utilized is resilience and mental resilience. Studies have shown their beneficial effect on nurses in terms of reducing COVID-19 pandemic burnout, job burnout, quiet quitting, and turnover intention [29,57]. In the case of workplace bullying, nurses with a high degree of resilience succeed in reducing the negative impact of bullying on the quality of their work life, as resilience acts as a mediating factor [58]. Although bullying negatively affects nurses’ self-efficacy, when it is cultivated and employed as the coping strategy for bullying, it serves as a mediating factor for the negative impact of bullying on both nurses’ mental health and on their intention to leave [59].

### Limitations

Our study had several limitations. First, we cannot infer a causal relationship between workplace bullying and quiet quitting since we employed a cross-sectional design. Longitudinal studies that prospectively measure workplace bullying, quiet quitting, and coping strategies among nurses will add significant information. Second, we conducted a web-based study and, therefore, we cannot calculate the response rate. Future studies should employ a paper–pencil survey method that allows scholars to calculate the response rate. Third, we used a convenience sample of nurses in Greece. There is no nurses’ registry in Greece and, thus, a random sample cannot be achieved. For instance, the majority of our nurses were females and have been working in understaffed departments. Thus, we cannot generalize our results although we achieved the minimum sample size for our study. Further studies with random and more representative samples would add invaluable knowledge. Additionally, studies in other countries with different clinical and cultural settings will offer the ability to make comparisons. Fourth, we examined the mediating effect of coping strategies on the relationship between workplace bullying and quiet quitting. Other variables can also act as mediators and should be investigated in the future. For instance, personality characteristics, organizational variables, and managers’ characteristics such as transformational leadership and organizational support can be considered as mediators in future studies. Fifth, we considered several socio-demographic characteristics as covariates in the mediation models. However, scholars should include more socio-demographic variables in the mediation models to further increase the validity of the results. Finally, we used self-reported scales to collect our data. Although our scales are valid and reliable, information bias is still possible.

## 5. Conclusions

Bullying is a negative aspect of the nurses’ work environment, with a significant prevalence. The present study highlighted its impact on nurses’ quiet quitting and the crucial role of bullying coping strategies in mediating its impact on quiet quitting. As quiet quitting is becoming increasingly prevalent across a wide range of businesses, including the healthcare sector, reducing the phenomenon of bullying should be an important priority for the management of healthcare organizations, with the objective of addressing the phenomenon of quiet quitting as well. The participants of the present study were found to apply positive bullying coping strategies, which is an effective attitude towards bullying and contributes to the reduction of its negative effects. Studies in the existing literature show that nurses often choose negative coping strategies, which have an adverse effect on both the work environment and on themselves personally (e.g., substance use and mental health). Therefore, it has become imperative to implement educational interventions to train nurses in the adoption of positive bullying management strategies. At the same time, the administrators of healthcare organizations should carry out a diagnostic audit in the working environment of nurses, in order to identify the existence of factors that favor the development of bullying behavior and to proceed immediately to the implementation of necessary interventions.

## Figures and Tables

**Figure 1 healthcare-12-00797-f001:**
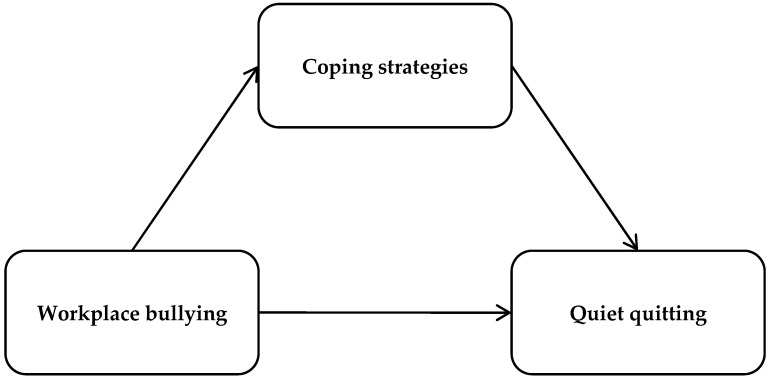
Structural model depicting the relationships between workplace bullying, coping strategies, and quiet quitting.

**Figure 2 healthcare-12-00797-f002:**
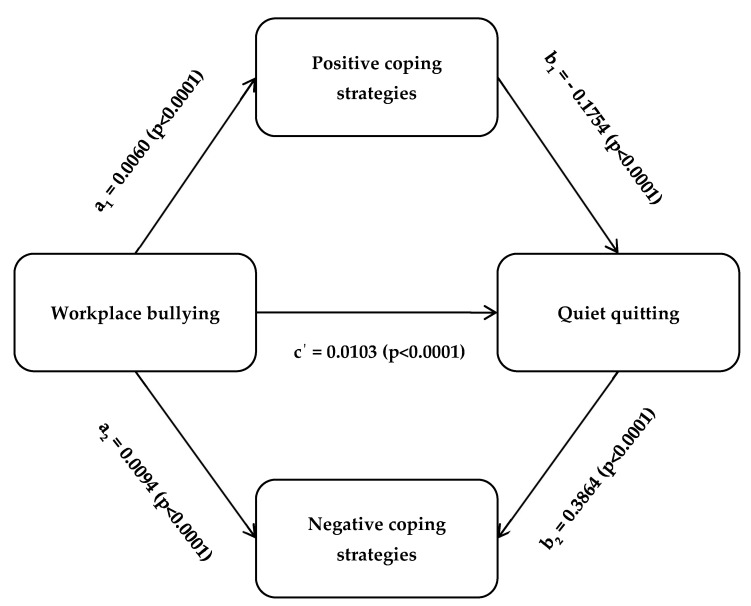
Structural mediation model using PROCESS macro with path coefficients and *p*-values of positive and negative coping strategies as the mediators in the relationship between workplace bullying and quiet quitting.

**Table 1 healthcare-12-00797-t001:** Demographic and job characteristics of nurses (N = 665).

Variables	N	%
Gender		
Males	82	12.3
Females	583	87.7
Age (years) ^a^	38.9	10.1
Understaffed department		
No	132	19.8
Yes	533	80.2
Clinical experience (years) ^a^	14.1	10.2
Shift work		
No	170	25.6
Yes	495	74.4

^a^ mean, standard deviation.

**Table 2 healthcare-12-00797-t002:** Descriptive statistics for the study scales (N = 665).

Scale	Mean	Standard Deviation	Median	Minimum Value	Maximum Value	Range
Negative Acts Questionnaire—Revised	51.3	20.6	46.0	22	108	86
Quiet Quitting Scale	2.5	0.6	2.4	1	5	4
Brief COPE						
Positive coping strategies	2.6	0.5	2.6	1	4	3
Negative coping strategies	2.0	0.5	2.0	1	4	3

**Table 3 healthcare-12-00797-t003:** Pearson’s correlation analysis among workplace bullying, quiet quitting, positive coping strategies, and negative coping strategies.

Variables	Quiet Quitting	Positive Coping Strategies	Negative Coping Strategies
Workplace bullying	0.453 ***	0.244 ***	0.423 ***
Quiet quitting		0.060	0.395 ***

*** *p* < 0.001.

**Table 4 healthcare-12-00797-t004:** Multivariable linear regression analysis with quiet quitting as the dependent variable.

Independent Variables	Coefficient Beta	95% Confidence Interval	*p*-Value
Males vs. females	0.156	0.041 to 0.271	0.008
Age	−0.001	−0.005 to 0.003	0.511
Understaffed department (yes vs. no)	0.033	−0.064 to 0.130	0.508
Shift work (yes vs. no)	0.076	−0.014 to 0.165	0.096
Workplace bullying	0.010	0.008 to 0.012	<0.001
Positive coping strategies	−0.169	−0.251 to −0.087	<0.001
Negative coping strategies	0.382	0.285 to 0.479	<0.001

**Table 5 healthcare-12-00797-t005:** Mediation effect of coping strategies on the relationship between workplace bullying and quiet quitting.

Outcome	Mediation Analysis Paths	Regression Coefficient	SE	95% Bias-Corrected CI		*p*-Value
				LLCI	ULCI	
Quiet quitting	Total effect	0.0128	0.0010	0.0109	0.0147	<0.0001
	Direct effect	0.0103	0.0010	0.0082	0.0122	<0.0001
	Indirect effect of positive coping strategies	−0.0011	0.0003	−0.0017	−0.0005	<0.0001
	Indirect effect of negative coping strategies	0.0036	0.0006	0.0084	0.0049	<0.0001
	Workplace bullying → Positive coping strategies	0.0060	0.0090	0.0042	0.0078	<0.0001
	Positive coping strategies → Quiet quitting	−0.1754	0.0418	−0.2574	−0.0933	<0.0001
	Workplace bullying → Negative coping strategies	0.0094	0.0008	0.0079	0.0110	<0.0001
	Negative coping strategies → Quiet quitting	0.3864	0.0496	0.2891	0.4837	<0.0001

LLCI: lower limit of confidence interval; SE: standard error; ULCI: upper limit of confidence interval.

## Data Availability

The data presented in this study are available upon request from the corresponding author.

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
