# Peer review of "Impact of Workplace Bullying on Quiet Quitting in Nurses: The Mediating Effect of Coping Strategies"

_healthcare, 2024, doi:10.3390/healthcare12070797_

Round 1

Reviewer 1 Report

Comments and Suggestions for Authors

Workplace bullying

Introduction – Comprehensive intro .  highlights organizational issues leading to bullying.  Positive and negative coping strategies of the recipient evaluated in terms of mediating bullying.  Defines quiet quitting.  Clearly explains two hypotheses of the study.

Materials/Methods and statistical analysis – clearly described.

Results – Tables 3 – 5 are a bit difficult to follow but with the written commentary I was able to match everything up.  Except for Table 3 – should there really be any correlation between positive and negative coping strategies?

Limitations – good summary of potential limitations

Discussion/Conclusion – Clear summation of the relationships between bullying, positive or negative coping strategies and resultant amount of quiet quitting.  Lines 272 – 292 again note the organizational issues that lead to bullying – the onus of remediating the bullying once again falls to the nurses (by teaching them positive coping strategies and then employing them leading to increased resilience) and to the nurse managers in providing a high relationship-oriented leadership style.  This study looked at coping strategies and resultant degree of quiet quitting; it did not study the underlying organizational issues.  Nevertheless – The need to look at and remediate the underlying organizational issues should be stressed in the Conclusion section.

Author Response

Dear Reviewer,

Thank you very much for the peer review of the paper “Impact of workplace bullying on quiet quitting in nurses: the mediating effect of coping strategies” and your comments, which have improved the quality of the manuscript.

Our best regards,

The Authors

Reviewer 2 Report

Comments and Suggestions for Authors

The article presents the important and timely topic of workplace bullying and its consequences, especially in the context of the quiet departure of nurses. The authors address an important issue that affects both employees and health care employers, with potentially harmful consequences for both parties. However, an analysis of the article reveals some shortcomings and deficiencies that may affect its overall value.

The introduction of the article requires some justification and better justification for the choice of topic. Despite the introductory discussion of the relevance of workplace bullying, there is no clear demonstration of the research gap that would be filled by the presented study. The introduction should also give more prominence to the contribution that the presented study makes to science, not only in terms of general awareness, but also in terms of practical implications for personnel management and health care organizational policies.

An important element that requires attention is the poorly described research process, including sampling. Is the study group representative of the industry in Greece? The lack of detailed discussion of this question weakens the reliability of the results and their possible generalizability. The article also lacks information on methodology, such as the adaptation to Greek conditions of the bullying measurement tools used, the criteria for determining quiet quits and also mediating variables. More clarity in this regard would benefit the reader.

The statistical analysis was done correctly, which is a strength of the article. However, the discussion of the results is limited, and reference to the existing literature on the subject is insufficient. The authors could have better integrated their results into the context of existing research and theory on workplace bullying, which would have helped readers understand the significance and implications of their findings.

The limitations section should be expanded to include items on future research. Pointing out gaps in research that remain to be filled and suggesting potential directions for development could enhance the article's value and its impact on further research in the field of workplace bullying.

In conclusion, the article touches on the important topic of workplace bullying and its impact on quiet quitting among nurses in Greece. However, the existing deficiencies in the description of the research process, the limited discussion of the results, and the insufficient reference to the existing literature are challenges that need to be overcome. Improving these elements could significantly enhance the value of the article and its contribution to the field of workplace bullying research.

Author Response

(The authors gave the same response as above.)
